# Cancer Survivors’ Long-Term Health Service Costs in Queensland, Australia: Results of a Population-Level Data Linkage Study (Cos-Q)

**DOI:** 10.3390/ijerph19159473

**Published:** 2022-08-02

**Authors:** Katharina M. D. Merollini, Louisa G. Gordon, Yiu M. Ho, Joanne F. Aitken, Michael G. Kimlin

**Affiliations:** 1School of Health and Behavioural Sciences, University of the Sunshine Coast, Maroochydore, QLD 4558, Australia; 2Sunshine Coast Health Institute, Birtinya, QLD 4575, Australia; 3Health Economics, Population Health Department, QIMR Berghofer Medical Research Institute, Herston, QLD 4006, Australia; louisa.gordon@qimrberghofer.edu.au; 4School of Public Health, The University of Queensland, Herston, QLD 4006, Australia; joanneaitken@cancerqld.org.au; 5School of Biomedical Sciences, Queensland University of Technology, St. Lucia, QLD 4072, Australia; m.kimlin@qut.edu.au; 6Rockhampton Hospital, Central Queensland Hospital and Health Service, Rockhampton, QLD 4700, Australia; yiuming.ho@health.qld.gov.au; 7Rural Clinical School, The University of Queensland, Rockhampton, QLD 4700, Australia; 8Cancer Council Queensland, Fortitude Valley, QLD 4006, Australia; 9School of Public Health and Social Work, Queensland University of Technology, Kelvin Grove, QLD 4006, Australia

**Keywords:** cancer survivors, health service use, costs and cost analysis, health economics

## Abstract

Worldwide, the number of cancer survivors is rapidly increasing. The aim of this study was to quantify long-term health service costs of cancer survivorship on a population level. The study cohort comprised residents of Queensland, Australia, diagnosed with a first primary malignancy between 1997 and 2015. Administrative databases were linked with cancer registry records to capture all health service utilization. Health service costs between 2013–2016 were analyzed using a bottom-up costing approach. The cumulative mean annual healthcare expenditure (2013–2016) for the cohort of N = 230,380 individuals was AU$3.66 billion. The highest costs were incurred by patients with a history of prostate (AU$538 m), breast (AU$496 m) or colorectal (AU$476 m) cancers. Costs by time since diagnosis were typically highest in the first year after diagnosis and decreased over time. Overall mean annual healthcare costs per person (2013–2016) were AU$15,889 (SD: AU$25,065) and highest costs per individual were for myeloma (AU$45,951), brain (AU$30,264) or liver cancer (AU$29,619) patients. Our results inform policy makers in Australia of the long-term health service costs of cancer survivors, provide data for economic evaluations and reinforce the benefits of investing in cancer prevention.

## 1. Introduction

Cancer survivorship is defined as the time from cancer diagnosis to the end of life. Different stages of survivorship have been described as acute (diagnosis to treatment), chronic (ongoing), long-term/late survivorship (≥5 years post diagnosis) and cured (disease-free) [1,2]. Worldwide, there are currently 19.3 million new cancer cases per year and this number is expected to increase to 28.4 million cases in 2040 [3]. The global burden of disease from cancer is substantial, with 10 million cancer deaths reported in 2019 and an estimated 250 million disability-adjusted life years (DALYs) due to cancer [4]. In Australia, there were an estimated 151,000 new cancer diagnoses in 2021, with over 1.1 million people with a cancer history currently alive. The 5-year relative survival of all cancers collectively is 70% [5]. Recent reports estimated that 8.8% of the total health expenditure in 2018/2019 was incurred by cancer and other neoplasms (or AU$11.7 billion) [6] whilst during 2015–16 it was AU$10.1 billion, including costs of cancer diagnosis, treatment, and national screening programs for colorectal, breast and cervical cancer [7].

Beside the enormous direct cost to the healthcare system, cancer diagnosis and treatment often have long-term health and financial costs for patients and their families. Complex late effects, both physical (e.g., fatigue, pulmonary, cognitive, neurological, secondary cancer, sexual and cardiac effects,) and psychosocial (e.g., anxiety, depression, fear of recurrence), require ongoing medical care and can substantially affect the quality of life of cancer survivors [8,9,10,11,12]. The indirect burden of cancer spans indirect economic costs, such as productivity losses, carer time, and reduced household income, as well as intangible costs that cannot be measured in monetary terms, such as disruption to family life and involuntary lifestyle changes [13,14,15].

Previous Australian studies on cancer-related expenditure are limited because they did not estimate long-term healthcare costs for all types of cancer on a population level beyond five years post diagnosis [16,17,18,19]. Research from other jurisdictions has focused on long-term costs for a specific type of cancer [20,21,22,23], or on a population-level up to seven years from diagnosis [24]. Others used a population or cost-of-illness approach to estimate the economic burden for individual types of cancer during a certain year, rather than long-term [25,26,27,28], or applied a model-based approach synthesizing published literature [29]. To date, there has been no population level research on the health service costs of cancer survivors beyond the initial years after treatment, including all age groups, based on detailed patient-level health service expenditure (reflecting gold standard bottom-up micro costing methodology [30]).

Cancer survivors are significantly more likely to be diagnosed with other chronic conditions that share similar risk factors, such as smoking, physical inactivity, overweight, poor diet, and increased alcohol consumption [31]. People with many co-morbidities have higher healthcare costs [32], and understanding these costs is important for cancer management services.

We investigated long-term health service use (i.e., up to 19 years after diagnosis) and associated costs for cancer survivors on a population level [33] from an Australian health service perspective. The aim of this study was to quantify long-term health service costs of cancer survivorship on a population level in Queensland, Australia. The study objectives were to estimate the most recent health service costs on a cohort and patient-level by type of cancer, age, vital status, type of health service and time since first primary cancer diagnosis.

## 2. Materials and Methods

### 2.1. Data Extraction & Linkage

A detailed description of data collection, data linkage, variables extracted (linkage variables/research variables) and data storage was provided in our Study Protocol manuscript [33] and is summarized below.

#### 2.1.1. Study Cohort

Queensland is the third most populous state in Australia, with over 5 million residents [34]. Every Queensland resident diagnosed with a first primary malignancy (excluding basal and squamous cell carcinoma of the skin), between January 1997 and December 2015 (over 19 years), was eligible for the study. Individuals were identified from the Queensland Cancer Register (QCR), which contains records of all Queensland residents diagnosed with cancer. Cancer is a legally notifiable condition in Queensland, except for basal and squamous cell skin carcinoma.

#### 2.1.2. Ethics Approval

Ethics approval was obtained from the University of the Sunshine Coast Human Research Ethics Committee (USC HREC Approval A/17/941) and from the Australian Institute of Health and Welfare (AIHW) Human Research Ethics Committee (EO2017/3/348). Approval for Queensland data extraction and linkage was obtained under the Public Health Act 2005 (grant RD007281).

#### 2.1.3. Healthcare in Australia and Data Linkage

Healthcare in Australia is provided by a mixture of private and public sector funding and relies on co-payment by private health insurance, or directly by the patient [35]. Medicare, a universal tax-funded health insurance system, was designed to provide affordable healthcare to citizens and permanent residents, including free public hospital treatment, subsidized allied health services and prescribed pharmaceuticals [36]. Primary care and specialist services provided out of hospital have independent fees, and for eligible services a set Medicare rebate is provided. Patients are often required to make an out-of-pocket (OOP) contribution, part of which may be covered by their private health insurance. For private patients admitted to public hospitals, around 75% of expenses are covered by Medicare [35].

Each Australian state, including Queensland, routinely records cancer diagnoses in population cancer registries and collects administrative data on hospital and emergency admissions. In order to capture healthcare services provided in primary, allied health, specialist and hospital care, our identified study cohort from the QCR was linked with national and state data from the Queensland Hospital Admitted Patient Data Collection (QHAPDC), Emergency Department Information System (EDIS), Healthcare Purchasing and System Performance (HPSP) data, Medicare Benefits Schedule (MBS) and Pharmaceutical Benefit Schedule (PBS). This allowed us to capture costs directly incurred by hospital or emergency admissions in public hospitals, as well as Medicare-subsidized primary or allied healthcare and prescription pharmaceuticals. MBS and PBS data also captured patient OOP claims data. This research does not include costs incurred by private healthcare provided without Medicare involvement, or funded by private health insurance.

Data linkage was conducted by the relevant state (Statistical Services Branch in Queensland Health) and national departments (AIHW) and transferred to a Secure Unified Research Environment (SURE) after de-identification of the data and assignment of a new, random patient ID; researchers obtained approval to remotely access de-identified records on the password-protected virtual platform SURE [33].

### 2.2. Data Preparation and Analyses

#### 2.2.1. Definition of Cancer Types

Cancer types were defined using topography codes, based on the World Health Organization’s International Classification of Diseases for Oncology, 3rd edition (ICD-O3) [37] and the 20 most common types of cancer in Australia were detailed in all subsequent analyses. The linked data was recoded according to these categories and the remaining cancer types were summarized as ‘all other cancers combined’.

#### 2.2.2. Cost Calculations

We focused on the most recent data available to ensure relevance of cost estimates, and selected healthcare costs incurred between 1 January 2013 and 31 December 2016. Individuals were included in cost calculations if they were diagnosed with a primary malignancy at any point from January 1997 to the end of December 2015, and incurred healthcare costs at some time between January 2013 and December 2016. All patient costs were included and patients who died incurred partial costs for that year. We used a bottom-up costing approach for the most accurate results. Individual-level patient records for each cost component were aggregated to form the total cost product, including hospital admissions, emergency presentations, medical and allied health services, and pharmaceuticals, as described below. Healthcare costs included all services utilized in the given timeframe, whether or not cancer related. All costs were reported in 2016 Australian dollars (AU$).

Hospitalization and Emergency Presentation costs: Healthcare purchasing data comprised individual cost per episode of care for both hospitalizations and emergency admissions between January 2013 and December 2016 and included total cost per episode, being the product of direct and overhead costs.

Medical and Allied Health Service costs: These data were extracted from the Medicare Benefit Scheme via Medicare claims records. Benefits paid represent the dollar value received by healthcare providers from the government. Patient out-of-pocket (OOP) costs were calculated as the difference between fees charged by providers and benefits paid. The sum of patient OOP costs and benefits paid made up total cost of Medical Services in our analyses.

Pharmaceutical costs: These data were extracted from the Pharmaceutical Benefit Scheme via claim records of drug prescriptions. Net benefit paid is the government contribution towards the cost of prescription medicinal products and, along with patient contribution, was used as the overall pharmaceutical cost.

Costs by time since diagnosis: Time since diagnosis was calculated as the interval between the most recent year of data collected, 2016, and year of cancer diagnosis (as per QCR records). Time since diagnosis was calculated in months and rounded to years, to form the following timeframes: 0–1 year, 2–4 years, 5–9 years, 10–14 years and 15–20 years since diagnosis. ‘Costs since diagnosis’ were calculated for all individuals with cost data incurred in 2016, excluding people who died that year.

Cost by type of cancer and relative 5-year survival: Types of cancer were categorized into cancer types with low, medium, or high relative 5-year survival rates. These were defined as: low for 5-year relative survival rate of 0–35% (i.e., pancreas, lung, liver, esophagus, brain, stomach, unknown primary site), medium for 36–69% survival rate (i.e., ovary, bladder, myeloma, leukemia, kidney) and high for 70–100% survival rate (i.e., colorectal, head & neck, cervix, non-Hodgkin lymphoma, uterus, breast, melanoma, prostate, thyroid) [38,39].

### 2.3. Data Cleaning and Statistical Analyses

Data were checked for missing values, logical errors and outliers, but no extreme values were excluded to represent the whole range of health service utilization. In the original QCR data, sex was only used as a linkage variable and was not available to researchers. We were able to access sex of the original patient cohort by merging data from QHAPDC, using gender-specific cancer codes (C51-C58: female cancers; C60-63: male cancers) and gender recorded in EDIS. Mean and median costs, cost ranges, standard deviations and cumulative costs were calculated for each cost component per patient per year, as well as per episode of care. SPSS software version 21.0 was used for all statistical analyses and data were illustrated graphically using Microsoft Office Excel version 2108.

## 3. Results

### 3.1. Patient Cohort Selection

A total of N = 365,443 individuals were ascertained from the Queensland Cancer Register (Figure 1). Of these, 130,102 (35.6%) had died prior to 2013 and a further 4961 (1.4%) were excluded from analysis because of missing healthcare cost data for the period 2013–2016. The final cohort for analysis comprised the remaining N = 230,380 individuals.

### 3.2. Demographic Overview of Cancer Cohort

Demographic information for the selected cancer cohort of N = 230,380 individuals is shown in Table 1. This included slightly more males than females (51.8% vs. 46.9%), with a small proportion aged 0–44 (14.3%), and most individuals aged 45 to 64 (42.5%) or over the age of 65 (43.2%). The most common geographic locations of birth (according to the Standard Australian Classification of Countries [40]) were Australia (76.9%), Northwest Europe (10.9%) and Oceania (4.9%). Many individuals in the cohort were married (66.1%) and the most common cancer types recorded were prostate cancer (17.6%), melanoma (17.5%), breast (16.2%) and colorectal cancer (11.5%), making up a total of 62.8% of all cancer diagnoses. Mean age at diagnosis was 60.5 years and mean age at death 74.7 years.

### 3.3. Mean Annual Healthcare Cost on Patient-Level

#### 3.3.1. Costs by Age Group and Vital Status

Mean annual healthcare costs, between 2013–2016, were AU$15,890 (SD 25,036) per person. Figure 2 illustrates these costs by age group and vital status. The red bars represent patients who were at the end of life and died during this timeframe and consistently showed higher costs compared to patients still alive. Costs were highest for the younger age groups (<20 years). Patients aged 10–14 years who had recently died had the highest mean annual costs of AU$121,273, followed by patients in age groups 5–9 years (AU$97,311), 0–5 years (AU$77,804) and 15–19 years (AU$71,327). Individuals who were alive during 2013–2016 (blue column) had the highest costs per patient for ages 0–4 years (AU$24,197), 5–9 years (AU$22,287) and 10–14 years (AU$20,993), followed by patients aged 75–84 years, incurring >AU$17,000 per year. More details on these costs can be found in Appendix A.

#### 3.3.2. Costs by Time since Diagnosis


Distribution of Cancer Cohort by Time since Diagnosis and Vital Status


Based on reference year 2016, the distribution of time since diagnosis by vital status is shown in Figure 3. Individuals diagnosed within the last year (prior to 2016) made up 10.1% of the cohort, with most patients diagnosed 2–4 years prior (27.9%), 5–9 years (30.1%) and 10–14 years ago (19.5%), with a smaller percentage diagnosed 15–20 years ago (12.4%). Most patients in the cohort of N = 230,380 with healthcare costs incurred during 2013–2016 were alive (88.3%) in 2016, and incidence of deaths was highest 2–4 years (5.5%) and 5–9 years (2.7%) after diagnosis.


Costs by Time since Diagnosis


Mean healthcare costs per person incurred in 2016 by time since diagnosis, hence only for individuals with vital status ‘alive’ in 2016, are shown in Figure 4 (N = 203,495). Healthcare costs were much higher in the first year (AU$23,896) compared to medium (2–4 years since diagnosis: AU$16,900) or long-term outcomes (5–20 years since diagnosis: AU$10,000–AU$11,000).

#### 3.3.3. Costs by Type of Cancer and Relative 5-Year Survival Group

Mean annual healthcare costs per person between 2013 and 2016 by type of cancer, and 5-year relative survival, are shown in Figure 5. Highest mean costs per individual for cancers with low survival cancers were incurred by individuals with a history of brain and liver cancer, with around AU$30,000 each; and in cancers with medium survival by myeloma (nearly AU$46,000) and leukemia patients (AU$29,158). In cancers with a high 5-year relative survival, the highest mean costs per year were incurred by non-Hodgkin lymphoma (AU$24,397). The lowest mean costs were recorded for individuals diagnosed with melanoma (AU$9487) and thyroid cancer (AU$8808). Overall, mean annual healthcare cost per individual for all cancers combined was AU$15,890 (SD: 25,036).

#### 3.3.4. Costs by Type of Cancer and Health Service Component

Figure 6 breaks these costs down further by health service component. Cancer types are sorted by highest mean annual healthcare costs per person (as indicated by blue line). It should be noted that these health service component costs are weighted mean costs based on individuals who received these services between 2013 and 2016, but these differed for each type of health service and each cancer type, e.g., not every individual experienced hospitalization or emergency admissions during this time frame. Detailed information for each of these components can be viewed in Appendix A, including the number of patients receiving each service and related costs. Hospitalizations were experienced by 48.6% of individuals between 2013 to 2016, and were the component with the highest mean patient cost per year for all types of cancer. Mean annual cost per person with at least one hospitalization were AU$26,431 (SD 31,246), compared to AU$5948 (SD 9672) for individuals without hospitalization. Mean annual hospitalization costs per person per year were AU$17,297 (SD 27,542), with the highest costs incurred by individuals with brain cancer (AU$29,873, SD 1199), leukemia (AU$28,543, SD 3395) and myeloma (AU$28,057, SD 1398). Costs for pharmaceuticals and Medicare services were incurred by nearly all patients (>97%) and emergency admissions by 47.9% of patients. Cancer types with the highest mean annual pharmaceutical costs per person per year were myeloma (AU$19,657, SD 20,772), leukemia (AU$8273, SD 15,130), non-Hodgkin lymphoma (AU$7063, SD 10,528) and liver cancer (AU$6036, SD 13,723), and all of these exceeded mean patient costs for Medicare services. Otherwise, Medicare services were higher than pharmaceutical costs and ranged from AU$2622 per person per year for thyroid cancer, to AU$7914 for myeloma. Mean annual patient costs for emergency admissions were comparatively low for all cancer types with AU$1471 (SD 1445), with a maximum of AU$2001 (SD 1793) for patients with a history of lung cancer.

### 3.4. Mean Annual Healthcare Cost on Cohort-Level

#### 3.4.1. By Age Group in Proportion to Total Cost

The total mean annual healthcare expenditure for the cancer cohort between 2013 and 2016 was AU$3.66 billion. Figure 7 displays the distribution of costs by age group and specifies costs in proportion to total costs and the total cancer cohort, and details can be found in Appendix A. The group with the highest total mean annual healthcare costs were adults aged 65–69 with AU$587.07 million spent, equivalent to 16% of the total cost per year. Most costs (73.9%) were accrued by individuals aged 50–79 years (69.9% of cohort). Younger age groups <40 years made up 9.4% of the total cancer cohort and 6.9% of total cost. Individuals diagnosed with cancer at <60 years of age showed proportionally lower costs compared to individuals >64 years old, making the cost distribution slightly left skewed. Nevertheless, the distribution of costs crudely aligns with the prevalence of cancer in our cohort.

#### 3.4.2. By Type of Cancer and Health Service Component

Total mean annual healthcare cost from 2013–2016 by type of cancer and health service component are illustrated in Figure 8. Overall, prostate cancer had the highest cost with AU$537.72 million, followed by breast cancer (AU$495.98 million) and colorectal cancer (AU$475.83 million); lowest cost cancers with <AU$32 million were esophageal cancer (AU$31.52m) and cervical cancer (AU$28.17m). Although total annual healthcare costs differ by type of cancer (as apparent in Figure 5), the distribution of cost by each health service category is similar for most cancer types. The mean proportion of costs per component are illustrated in Figure 9. The highest costs can be attributed to hospitalizations with 53% of total annual costs, followed by Medicare services with 24%, pharmaceuticals with 19% and emergency admissions with 4%. Myeloma had a higher proportion of pharmaceutical costs (42%), followed by hospitalizations (39%), Medicare services (17%) and emergency admissions (2%). Similarly, Non-Hodgkin lymphoma and leukemia had higher than average pharmaceutical costs (28%), whereas lung and brain cancer utilized higher than average hospitalizations (64%).

## 4. Discussion

### 4.1. Main Findings in Context of Recent Literature

Our cost analysis shows mean annual healthcare expenditure by individuals with a history of cancer of over AU$3.66 billion for the period 2013–2016, with the highest mean annual costs per person incurred by those with a history of myeloma (AU$45,951), brain cancer (AU$ 30,264), liver cancer (AU$29,619) or leukemia (AU$29,158). The most recent AIHW report on health system disease expenditure for Australia (2018–19), reported the highest expenditure from non-melanoma skin cancer (NMSC) (AU$1326m), breast cancer (AU$1313m), and prostate cancer (AU$1200m) [6], excluding benign, in situ and uncertain (non-malignant) neoplasms. They found that 8.8% of the total expenditure was due to cancer and other neoplasms, of which 58% of total costs were due to cancer care in hospitals, followed by pharmaceuticals (26.5%) and other referred medical and primary health care services (15.5%) [6]. Although we focused on malignant neoplasms and excluded NMSC, the percentage of costs for hospitalizations in public hospitals were similar in our data (53% of total expenditure), but in our data these were followed by Medicare services (24%) and pharmaceuticals making up around 19% of total annual healthcare expenditure. This difference may be partially explained by the fact that our analysis did not include cancer screening or costs relating to the cancer diagnosis, but started with costs incurred following diagnosis. Also, the AIHW estimates applied a combination of ‘top-down’ and ‘bottom-up’ approaches to estimate costs, including national data sources and private hospital data combined into a ‘Disease Expenditure Database’, which was used to model their estimates and may have resulted in higher overall expenditure [7]. Total healthcare costs for myeloma were proportionally higher than average pharmaceutical costs, compared to other types of cancer (42% vs. average 19%). These findings reflect treatment practices of patients requiring ongoing drugs to treat adverse effects, such as bisphosphonates, antibiotics and antiviral drugs [41].

In comparison, another Australian study by Goldsbury et al. used a different approach and estimated excess healthcare costs attributable to cancer (using a control group), which during 2013 was around $AU6 billion Australia-wide (based on an individual-level patient sample of n = 7624 participants in New South Wales diagnosed between 2009–13, aged 45 and older) [19]. Their estimate was also based on hospitalizations, emergency presentations, pharmaceuticals, and Medicare services. A main difference in cost estimates is that individual-level hospital and emergency admission costs were derived from patients’ diagnosis codes, linked to average costs as the Australian Hospital Cost data collection (AHCDC), whereas our costs had precise episode costs attached to each hospitalization/emergency admission, and hence are expected to be more accurate. They reported mean annual costs for the initial treatment phase of AU$28,719 (1–2 years post diagnosis), AU$4474 for the continuing phase (time after initial phase, before terminal phase) and AU$49,733 for the terminal phase (last year of life) [19]. We stated costs by time since diagnosis, rather than phase of care, which makes direct comparison of results challenging; our results showed mean annual health system expenditure per individual with cancer history was AU$15,890 across all types of cancer, with AU$23,896 in the first year, AU$16,900 between 2–4 years and slightly above AU$10,000 for 5–20 years since diagnosis (based on healthcare expenditure in 2016). Nevertheless, the expenditure of individuals with vital status ‘deceased’ showed consistently higher costs across all age groups, with highest mean annual costs for patients aged 10–14 years (AU$121,273).

Bates et al. published total healthcare system costs in Queensland for all cancers during the first 12 months after diagnosis (2011–12) of AU$4.8 billion [42]. These annual results are expected to be higher than our estimate of AU$3.66 billion given that they focused on the first-year post-diagnosis, where patients receive a substantial amount of treatment and healthcare interaction and, hence, are higher than costs for patients with a longer time since diagnosis. Our data showed that healthcare costs were substantially higher in the first year compared to patients between 2–4 years post diagnosis (~AU$2400 vs. AU$16,900), or beyond 5 years since diagnosis (5–20 yrs post diagnosis: >AU$10,000).

In the Canadian setting, De Oliveira et al. appraised the economic burden of cancer on a population level in the state of Ontario of CAD$7.5 billion in 2012 [24], and high costs for brain cancer and myeloma were also reported [43]. Their research and other work from Japan confirmed hospitalizations to be the highest cost component for healthcare resource utilization in cancer patients, as described here [24,28]. An estimate of cancer-related versus non-cancer related health care costs was provided by Sam et al. in 2019, again for the Canadian setting where hematologic and lung cancers were the costliest [44]. A recent systematic review by Essue et al. explored the psychosocial cost burden of cancer in Canada including psychological, physical, and spiritual dimensions, which have been described to range from CAD$427,753–528,769 per person over a lifetime [45]. They concluded that two thirds of economic costs are incurred by psychosocial costs and should be included in economic evaluations, but that there is currently a lack of a consistent measurement method [45]. In the European setting, a very comprehensive cost analysis on the economic burden of cancer in the European Union was published nearly a decade ago and provided estimates for direct healthcare costs and indirect costs incurred in 2009 [46]. They attributed 40% of costs to health care (€51 billion), with other costs being productivity losses due to early mortality (€42.6 billion), reduced working hours (€9.4 billion) and informal care (€23.2 billion) [46].

Although the costs by age group crudely aligned with the prevalence in the cohort, we found that younger individuals aged 0–14 years incurred the highest patient-level costs, although these only made up a small proportion of overall costs due to the small number of individuals. Healthcare expenditure for patients with vital status ‘deceased’ was the highest for 0–19-year-olds. This is likely to be due in large part to the very different mix of cancers that occur in children and adolescents, compared to older people. Leukemias (with one of the highest per person costs) make up almost a quarter of cancers diagnosed in people under 20 years, while breast, prostate, colorectal cancer and melanoma (with some of the lowest per person costs), make up a relatively higher proportion of cancers in older people. In Australia, there is some evidence of the use of expensive cancer drugs at the end of life, despite limited benefit [47]. This is partially due to socio-cultural and systemic factors, such as public reimbursement, but also societal attitude towards death and continuing with futile treatments at end of life [47]. More research is needed to explore our findings for young cancer patients.

Patient-level healthcare costs over time were nearly twice as high in the first year after diagnosis, compared to medium to long-term. Our data included around 10% of individuals in the first year after diagnosis, 58% between 2–9 years, and over 30% were long-term survivors of 10–20 years. Most deaths were recorded 2–5 years after diagnosis, which may have been attributable to a large proportion of individuals in the category. Given that costs drastically decrease after the first year, it is plausible that our cohort costs were lower compared to previous reports (as discussed above).

### 4.2. Strengths and Limitations

Limitations of this work were the use of health service data, which was created for administrative purposes rather than research purposes, and not all private hospitalizations were included in this data.

Also, the health service data included all types of costs incurred by any health service contact, rather than cancer-specific costs, and due to financial limitations we were unable to include a control group to calculate excess costs due to cancer. Details of cancer diagnoses were retrieved from the Queensland Cancer Registry data, which does not record basal and squamous cell carcinoma or stages of cancer; hence, these were missing from our analyses. Non-melanoma skin cancer (NMSC) cases were only reported occasionally, and costs data reported here should only be seen as an indication of costs per person, but not as a representation of overall costs for NMSC in this cohort. We only analyzed data for patients with available system cost data between 2013–2016, which means many patients from the initial cancer cohort were excluded; nevertheless, we are confident that the distribution of gender and types of cancer is nearly identical between the initial cohort, including all cases, and the latter cohort, as our cohort comparison has shown (details available on request). Nearly 3% of our selected cohort, with system cost data recorded between 2013–2016, showed a date of death prior to the date of health service contact, indicating deceased people receiving health care services. These seemingly implausible scenarios might be explained by government data (MBS/PBS) using ‘data perturbation’ to ensure non-identifiability of individuals, where the date of birth/death in the data is randomly selected from the actual date +/−3 months. As we were unable to account for this intricacy, we had to exclude this small proportion of people from further analysis. Furthermore, the focus of this work was on direct health services costs, rather than overall economic costs. Hence, not all relevant costs were included, such as opportunity costs, productivity costs or carer costs. Also, we only partially included patient out-of-pocket expenditure, reflected in co-contributions to prescribed pharmaceuticals (not available for over-the-counter drugs) and some Medicare services. Given that our data included population data, it was to be expected that some cases showed unusually high costs, but to reflect the full spectrum of health services costs, no cost outliers were excluded. Costs were reported in 2016 Australian dollars and not adjusted for the Consumer Price Index, due to the current instability of the inflation rate.

Strengths of this project comprise the use of highly reliable, routinely collected administrative healthcare data from a variety of sources, the inclusion of population data without any limitations on age or type of cancer, starting with the date of cancer diagnosis, and continuous records with long-term outcomes of up to two decades. Given that our sample is very large and population-based, our results are likely to be generalizable to the rest of the Australian population. Furthermore, we applied a bottom-up costing approach to estimate healthcare costs by summarizing individual costs, which is seen as the gold standard in healthcare costing, and to our best knowledge we were one of the first Australian studies to use population data to estimate health service costs of cancer survivors.

Future research on long-term cancer survivorship could explore health service costs in other settings on a population level, including developing countries and jurisdictions with different healthcare systems, as well as broader economic costs on a societal level.

## 5. Conclusions

We captured the whole journey of health service contact and were able to estimate related costs of cancer patients diagnosed and treated in Queensland over a period of 20 years. Our research outcomes form a body of evidence to show potential lifetime cost savings by cancer prevention and will provide a rich data source for future economic evaluations. We also inform policy makers in Queensland/Australia and hence facilitate forthcoming planning on the utilization of healthcare resources, according to the burden of disease, which will potentially lead to more investments in cancer prevention and/or survivorship care.

## Figures and Tables

**Figure 1 ijerph-19-09473-f001:**
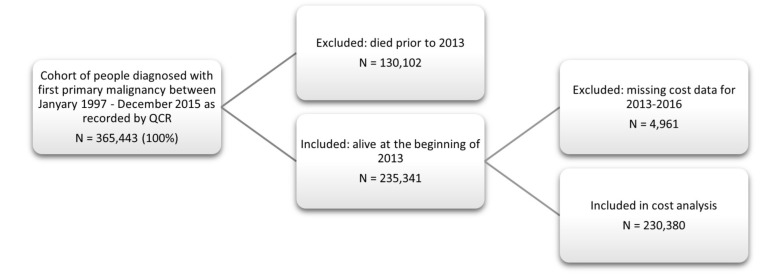
Flowchart for Selection of Queensland Cancer Cohort.

**Figure 2 ijerph-19-09473-f002:**
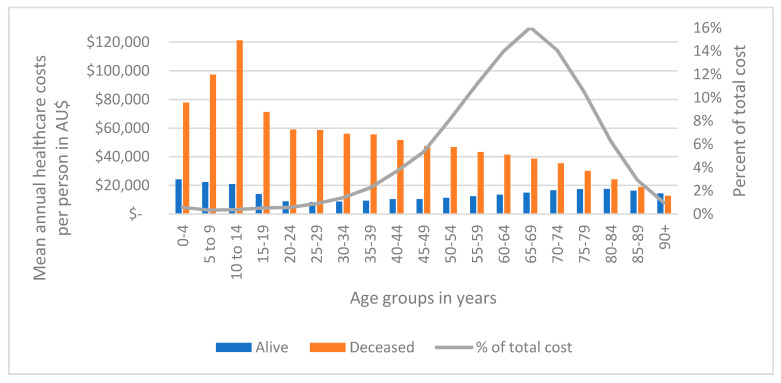
Mean annual healthcare costs (2013–2016) per person in AU$ by age and vital status (N = 230,380).

**Figure 3 ijerph-19-09473-f003:**
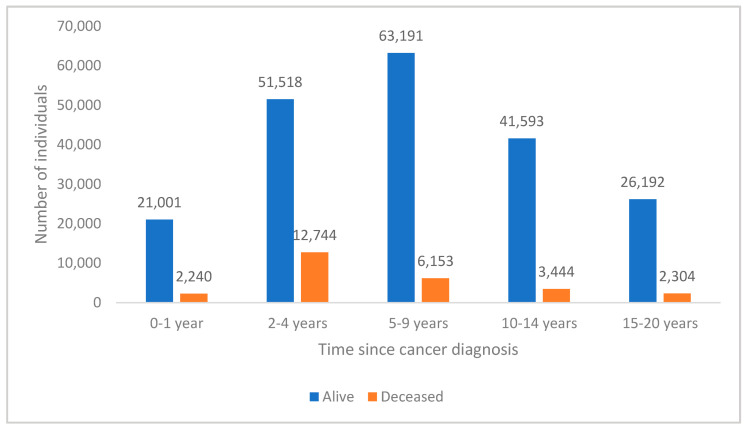
Cancer cohort distribution by time since diagnosis and vital status (N = 230,380).

**Figure 4 ijerph-19-09473-f004:**
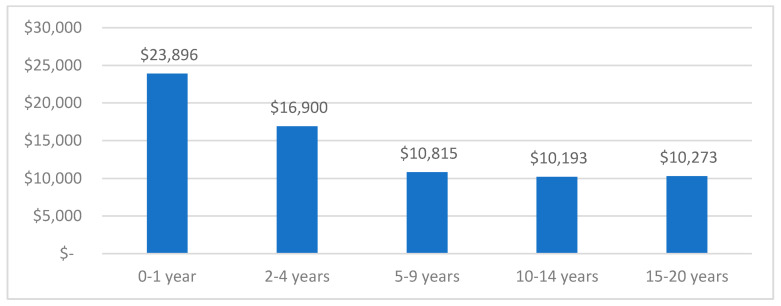
Mean healthcare costs (2016) per person in AU$ with vital status alive in 2016 by time since diagnosis (N = 203,495).

**Figure 5 ijerph-19-09473-f005:**
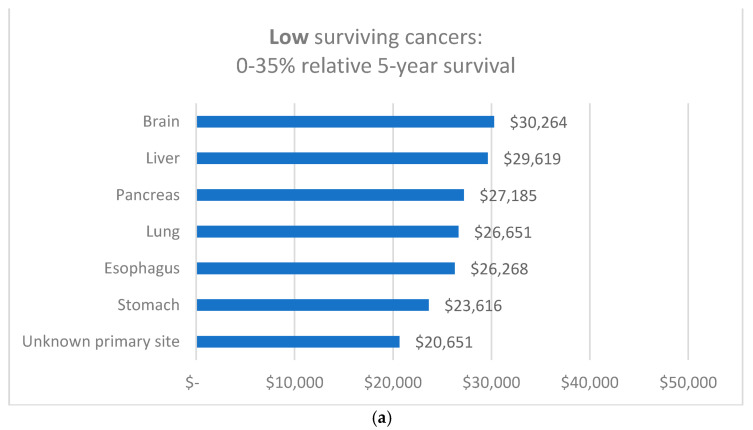
Mean annual healthcare costs in AU$ by (**a**) low, (**b**) medium and (**c**) high relative 5-year survival (N = 230,380).

**Figure 6 ijerph-19-09473-f006:**
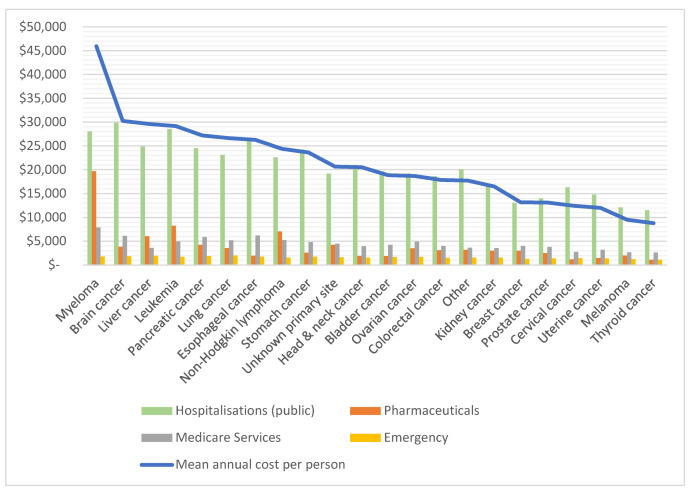
Mean annual healthcare costs per person (2013–2016) in AU$ by cancer type and health service component.

**Figure 7 ijerph-19-09473-f007:**
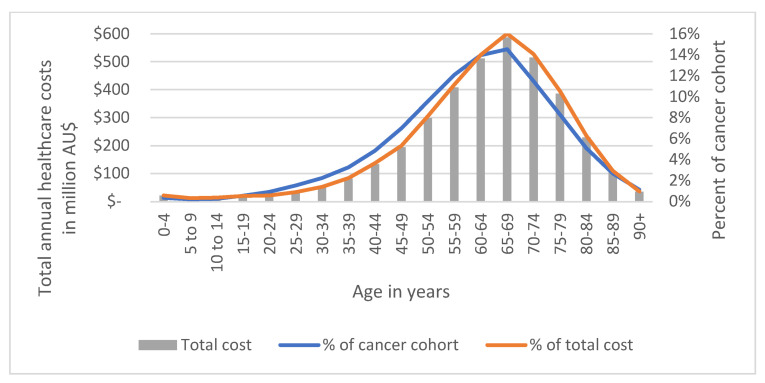
Total annual healthcare costs (2013–2016) in AU$ by age group and proportion of cancer cohort (N = 230,380).

**Figure 8 ijerph-19-09473-f008:**
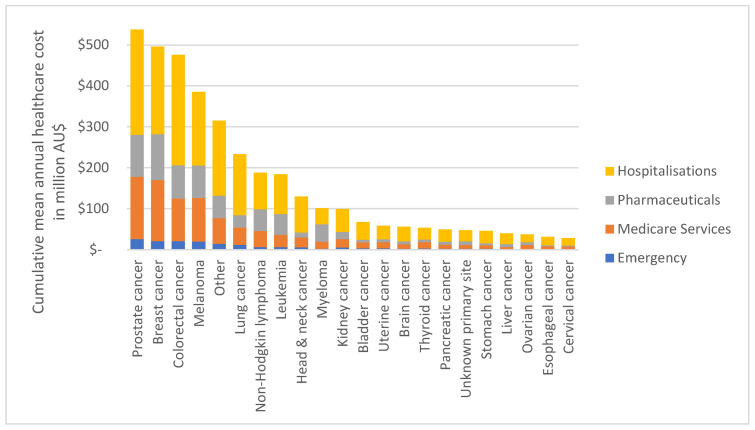
Cumulative mean annual healthcare cost (2013–2016) in AU$ for the cancer cohort by type of cancer and health service component (N = 230,380).

**Figure 9 ijerph-19-09473-f009:**
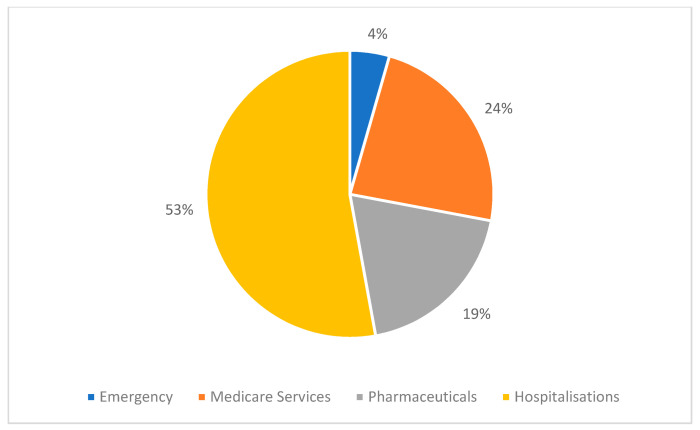
Proportion of health service component contributing to total mean annual healthcare costs (2013–2016) in AU$ for all types of cancer for patient cohort of N = 230,380.

**Table 1 ijerph-19-09473-t001:** Demographics of Queensland Cancer Cohort at time of first primary malignancy diagnosis (1997–2015) with healthcare utilization between 2013–2016.

Characteristics	Total (N)	Percent (%)
Total number of individuals	230,380	100
Gender ¹		
Male	119,137	51.8
Female	105,062	46.9
Age at diagnosis in years	N	%
0–14	1905	0.9
15–24	3383	1.4
25–34	8708	3.7
35–44	18,708	8.1
45–54	38,068	16.5
55–64	60,018	26.1
65–74	59,979	26.0
75–84	30,872	13.4
85+	8739	3.8
Mean age (SD)	60.5 (15.4)	
Geographic location of birth	N	%
Australia	176,994	76.8
Northwest Europe	25,210	10.9
Oceania except Australia	11,260	4.9
Southeast Europe	5611	2.4
Southeast Asia	2513	1.1
Sub Saharan Africa	2108	0.9
America	1814	0.8
Northeast Asia	1519	0.7
South and Central Asia	903	0.4
North Africa and Middle East	654	0.3
Other	1794	0.8
Marital status		
Married/De facto	152,427	66.2
Widowed	26,181	11.4
Divorced/Separated	25,580	11.1
Never married	22,521	9.8
Unknown	3671	1.6
Type of cancer		
Prostate	40,988	17.8
Melanoma	40,655	17.6
Breast	37,745	16.4
Colorectal	26,646	11.6
Lung	8750	3.8
Non-Hodgkin lymphoma	7700	3.3
Head & neck	6323	2.7
Leukemia	6308	2.7
Thyroid	6098	2.6
Kidney	6016	2.6
Uterine	4893	2.1
Bladder	3574	1.6
Unknown primary site	2303	1.0
Cervical	2266	1.0
Myeloma	2201	1.0
Ovarian	1997	0.9
Stomach	1941	0.8
Brain	1849	0.8
Pancreatic	1819	0.8
Liver	1342	0.6
Esophageal	1200	0.5
All other cancers combined	17,766	7.7
Survivorship status by 2016		
Alive	203,495	88.3%
Deceased (between 2013–15)	26,885	11.7%
Age at death		
Mean (SD)	74.7 (13.5)	

¹ Due to data linkage, there were a total of 6181 people with missing data on gender.

## Data Availability

The data that support the findings of this study are available from the data custodians of each of the linked datasets, but restrictions apply to the availability of these data, which were used under license for the current study, and so are not publicly available. Aggregated data used for this manuscript are, however, available from the authors upon reasonable request.

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
