# Peer review of "Cancer Survivors’ Long-Term Health Service Costs in Queensland, Australia: Results of a Population-Level Data Linkage Study (Cos-Q)"

_ijerph, 2022, doi:10.3390/ijerph19159473_

Round 1

Reviewer 1 Report

The paper analyzes the long-term health service costs of cancer survivorship on a population level. Using the case of Australia with patients diagnosed with a first primary malignancy between 1997 and 2015, the health service costs between 2013 to 2016 were analyzed using a bottom-up costing approach. The findings recommend policies reinforcing the benefits of investing in cancer prevention.

In general, the paper is well-written with a detailed methodology and well-discussed results. Only minor issues need to be addressed particularly the significance of the study from a broader perspective. Specifically,

1. Justify why Australia is the most appropriate (or best) case to analyze the cancer survivorship on a population level and not other countries.

2. In the Introduction, state the specific research objectives/questions.

3. Discuss briefly the policy implications of the main findings.

4. Discuss the broader implications of the findings such as the investments in health care systems in developing countries with little to no enough government support for cancer patients.

Author Response

Dear reviewer,

Thank you very much for your time and efforts in reviewing our manuscript.

Please see below for our detailed responses to your suggestions and changes made to the manuscript in blue/green font.

Kind regards,

Katharina Merollini (on behalf of all authors)

Reviewer 2 Report

With great interest I have read the work of Merollini et al. on cancer survivors’ long-term health service costs in Queensland, Australia: results of a population-level data linkage study (COS-Q).

The aim of this work was to quantify long-term health service costs of cancer survivorship on a population level, comprising the period between 1997 and 2015.

Authors should be congratulated on their great work.

Comments and Suggestions for Authors

1. The strengths of the manuscript are: the use and linkage of large administrative databases to estimate the total yearly cost of cancer and cost per year after cancer first diagnosis organized in age and cancer groups.

2. The perceived limitations are: data are primarily created for administrative purposes. Private hospitalizations and out-of-pocket expenditure is missed. Costs are not adapted for inflation in the observed period, over years.

3. The title accurately conveys the message of the paper.

4. The abstract/summary is a faithful outline of the paper and can be understood without reading the manuscript. No discrepancies exist between the abstract and the remainder of the manuscript.

5. The introduction succinctly lay the groundwork for what was done and the justification for this study.

6. Materials and Methods
The hypothesis and the aims are clearly stated.
The study design is appropriate to allow the hypothesis to be tested. The methods are clear.

7. Results
The results are valid based on the methods used. Results are presented as discussed in the methods.

8. Discussion
The discussion adequately compares and contrasts the results with those of other papers that have previously been published. Study limitations are appropriately addressed. The conclusion is valid.

9. Figures and tables are appropriate. References are appropriate, current and comprehensive.

I would recommend the following adaptations:

-        Lines 71-79: Please reformulate the last paragraph of the introduction to focus mostly on your goal / aim of the study, avoiding discussion here. This paragraph should clearly tell the reader what is the aim of your work.

-        Why did you decide to focus only on the period from 2013-2016? And not the whole period? Please describe in methods the reason for selection of this period.

-        Please describe the method used to linkage the data from different databases.

-        Please describe statistics in more detail, as it is usually. Which test did you perform for which analysis, and in dependence of data normality.  

-        Line 218: it was unknown gender, but this should be rather expressed as “with missing data on gender”.

-        Line 222: To make this easier to follow, I recommend reporting on SD as follows: AU$15,890 ±25,036. Reconsider this also for the abstract and the rest of the text.  

-        Did you adjust the prices for the Consumer Price Index (CPI)? If not please report this in limitations, or perform this adjustment.

-        Section 4.2: Please report also on out-of-pocket costs that are later not claimed by the patient but occurred.

-        Line 472: remove the word furthermore, it is repeated in two sentences. 

Author Response

Dear reviewer,

Thank you very much for your time and efforts in reviewing our manuscript and your excellent comments and suggestions for improving our manuscript.

Please see below for our detailed responses to your suggestions and changes made to the manuscript in blue/green font.

Kind regards,

Katharina Merollini (on behalf of all authors)

Reviewer 3 Report

This manuscript indicated long-term health servise costs of cancer survivor. It is very important for economic evaluations and investigation of cancer prevention. But, it may be required for some corrections.  Some figures showed menas, but not SD or SEM and statistical significant. It was better to add statistical analysis for  maltiple comparisons. 

  1) Major issues: What is financial burden for patients, their families and their carers predicted from this study ?

It is better to add a comparison between Research specific age groups (childhood, adolescent and young adult, adult and/or elderly survivors) and total age. 2) Minor issues: Quality of each figure and table is low. It is better to add some background in detail in the Introduction section.  

Author Response

(The authors gave the same response as above.)

Round 2

Reviewer 3 Report

This manuscript was corrected clearly.